# Research on improved gangue target detection algorithm based on Yolov8s

**Zhibo Fu**[1]*, **Xinpeng Yuan**[1]*, **Zhengkun Xie**[1], **RunZhi Li**[2], **Li Huang**[1]

**1** School of Coal Engineering, Shanxi Datong University, Datong, China, **2** Shenyang Research Institute, China Coal Technology and Engineering Group, Shenyang, China

* fzb20000416@163.com (ZF); XpnYuan@163.com (XY)

## Abstract

An improved algorithm based on Yolov8s is proposed to address the slower speed, higher number of parameters, and larger computational cost of deep learning in coal gangue target detection. A lightweight network, Fasternet, is used as the backbone to increase the speed of object detection and reduce the model complexity. By replacing Slimneck with the C2F part in the HEAD module, the aim is to reduce model complexity and improve detection accuracy. The detection accuracy is effectively improved by replacing the Detect layer with Detect-DyHead. The introduction of DIoU loss function instead of CIoU loss function and the combination of BAM block attention mechanism makes the model pay more attention to critical features, which further improves the detection performance. The results show that the improved model compresses the storage size of the model by 28%, reduces the number of parameters by 28.8%, reduces the computational effort by 34.8%, and improves the detection accuracy by 2.5% compared to the original model. The Yolov8s-change model provides a fast, real-time and efficient detection solution for gangue sorting. This provides a strong support for the intelligent sorting of coal gangue.

## 1. Introduction

Due to the special energy structure of China, coal will still be the main consumed energy in the long-term future [1]. However, the process of coal mining produces a large amount of coal gangue, and this waste has become one of the critical solid pollution sources that endanger the human living environment [2]. As a result, the classification and treatment of coal gangue has become one of the important issues to promote the clean use of coal, and has received increasing attention. Coal banding identification has been widely used as an essential part of intelligent classification, density identification method, hardness identification method, ray identification method, and image identification method based on density, hardness, grayscale, and texture characteristics. Deep learning technology identification gangue devices are simple, easy to implement, green, and other advantages. This is an influential approach to achieve intelligent gang classification. Compared with these traditional methods, deep learning technology has become the mainstream of the current development of coal gangue recognition technology [3, 4] and has achieved significant results in the field of coal gangue recognition.

**Data Availability Statement:** All relevant data for this study are publicly available from the GitHub repository (https://github.com/zhibofu).

**Funding:** 1.Science and Technology Innovation Program for Higher Education Institutions in

Shanxi Province 2.Teaching Reform and Innovation Program for Higher Education Institutions in Shanxi Province (General Program) 3.Shanxi Datong University Graduate Education Reform Research Program 4.Shanxi Datong University Graduate Education Innovation Program.

**Competing interests:** The authors have declared that no competing interests exist.

As the coal industry gradually moves in the direction of digitization, networking and intelligence, research on gang identification is growing. Zhang [5] and his team proposed a gangue sorting system based on the γ-ray dual-energy projection method [6], which uses γ-ray sources of different energies as an excitation, and realises the accurate identification of gangue by irradiating the intensity of radiation flux produced by the gangue. Yang [7] and his team also proposed a method for gangue identification and sorting using dual-energy γ-rays by projection. He [8], on the other hand, used X-ray projection to identify gangue by setting a threshold based on the energy attenuation of X-rays by coal and gangue and identifying gangue by comparing the energy attenuation of X-rays by the target at the time of detection with the threshold. However, there are some drawbacks of the gangue sorting system based on the radiometric method, including elevated cost and long-term maintenance costs, and the possible threat of the radiometric detectors to the health of the workers [9]. The vibration detection method is a method to differentiate between coal and rock by establishing signal acquisition on the body of the coal mining machine or on the top and bottom of the rock to cut the coal and rock seams with different frequencies and vibration amplitudes [10]. Traditional methods of vibration detection include mechanical sensors, electrical sensors and hydraulic systems, among other forms. Although vibration signals have the advantages of easy detection, strong anti-jamming capability, and convenient transmission, they are not conducive to the realization of accurate mining, cause greater damage to coal mining machines, and require positional changes during coal mining that affect equipment maintenance. Radar detection, THz detection and electron spin resonance methods are commonly used electromagnetic detection techniques. These methods are based on the differences in the physical properties of coal and rock, and use various properties such as the speed, time, phase and return loss rate of electromagnetic waves as they propagate through coal and rock layers to achieve accurate identification of coal and rock. Among them, the radar detection method is characterized by its wide detection range, great accuracy, and strong anti-jamming capability, and thus has become one of the most commonly used identification methods. The THz sounding method applies the analysis of time delays and decay amplitudes to provide insights into data spanning different rock formations. The electron spin resonance method is a technique that uses electromagnetic waves emitted by coils and antennas to resonate, and it is a method that realizes the effect of the magnetic field created by measuring and evaluating the power of the received signal. Electromagnetic detection can detect information about cracks, fissures and other imperfections present in the interior of coal rock bodies, providing effective data for underground gas extraction in coal mines. However, due to the restriction of the characteristics of coal, the electromagnetic detection method can not be widely used because of the great number of interfering factors on the signal, the rapid signal attenuation speed, and the considerably reduced accuracy [11, 12]. Minghui Zhao [13] et al. proposed a method to identify coal gangue using CornerNet-Squeeze model to effectively reduce the interference of conveyor belt background in target detection. On the other hand, Cao Xiangang [14] et al. proposed a deep learning based technique for recognising gangue images, which improves the training effect and accuracy by migration learning and sharing the weights and biases of the convolutional layers of the trained model. Different scholars have also proposed various improved network models, such as the improved method based on the module structure of MobileNetV3-large [15], the improved Yolov3-M [16] model, the deep learning network based on Yolov4 [17], the improved Yolov5 model [18–21] and Yolov8 model [22]. Although these methods have improved the accuracy of gang recognition to varying degrees, they still have some drawbacks. Such as the amount of model parameters becomes additive, the running time becomes longer, etc. Taking CIoU_Loss as an example, the method is mainly used to optimise the detection by calculating the overlap area between the detection frame and the real frame, but it can not be well applied when there is a

containment phenomenon between the detection frame and the real frame, and the loss function itself is slow to converge in the horizontal and vertical directions, which is not adequate to meet the demand for the sorting of coal gangue.

Therefore, improving on the Yolov8s model by replacing the Yolov8s backbone network with Fasternet and introducing Slimneck in the head module of the backbone network accomplishes an excellent trade-off between model accuracy and speed, increasing the convergence speed, decreasing the number of parameters and calculations in the model, and increasing the detection rate. Faster convergence and more accurate regression results are obtained by using DIoU for better coal gangue differentiation. Finally, DyHead is used to significantly improve the expressive power of the object detection head of the model. This paper is divided into the following sections: Yolov8 algorithm briefly introduces the development of Yolo algorithm and introduces Yolov8; Improvement of the model focuses on the improvement method used in this paper; Experiments and results focuses on the experimental results and comparative experiments; Conclusion gives the conclusions and the direction of improvement.

## 2. Yolov8 algorithm

You-Only-Look-Once (YOLO) algorithm, which segments the image into multiple grids, predicts the bounding boxes within each grid and the classes of objects they contain, and eliminates the overlapping bounding boxes using a non-maximal suppression (NMS) algorithm. Yolov1 [23] has a relatively rapid detection speed, but it is not effective for objects that are close to each other and smaller targets. The Yolov2 [24] algorithm uses Darknet19 as a feature extraction network, which is able to adapt to images of different sizes and improve the detection accuracy for miniature targets. Yolov3 [25] added Feature Pyramid (FPN) and Spatial Pooling Pyramid (SPP) modules to improve the ability to detect different scales and semantic information. Yolov4 [26] introduced the Mish activation function to improve the accuracy. Yolov5 [27] introduced the C3 module and the SPPF module to improve the feature perturbation and enhance the detection capability. Yolov7 [28] achieves additional optimisation in object detection by employing a scalable and efficient layer aggregation network, E-ELAN, an innovative transition module, and a reparametrized structure strategy that enhances feature extraction and semantic information representation.

The YOLO algorithm, one of the classical single-stage detection algorithms, has been upgraded to Yolov8. One of the key features of the Yolov8 algorithm, the newest model in the YOLO family released by Ultralytics in 2023, is its scalability. It is designed as a framework that supports all previous versions of YOLO, allowing easy switching between versions and comparing their performance [29]. In addition to scalability, Yolov8 includes a number of different innovations that make it useful for a wide range of object detection and image segmentation tasks. These include a different backbone network, a different unanchored network detection head, and different loss function features. Yolov8 is also highly efficient, enabling operation from CPU to GPU. The backbone of Yolov8 is fundamentally the same as that of Yolov5, with the C3 module replaced by the C2f module based on the CSP idea. The C2f module borrows the idea of ELAN from Yolov7, and C3 and ELAN are combined to form the C2f module, which enables Yolov8 to obtain richer gradient flow information while ensuring its own light weight. At the end of the backbone, the most popular SPPF module [30] is still used to pass three Maxpools of size $5 \times 5$ in turn, and then each layer is connected in series, which ensures the accuracy of the object at different scales and the lightness of the object at the same time. In the neck part, the feature fusion method used by Yolov8 is still PANFPN [31], which enhances the fusion and utilisation of feature layer information at different scales. The authors of Yolov8 used two upsampling and multiple C2f modules as well as a final decoupled head

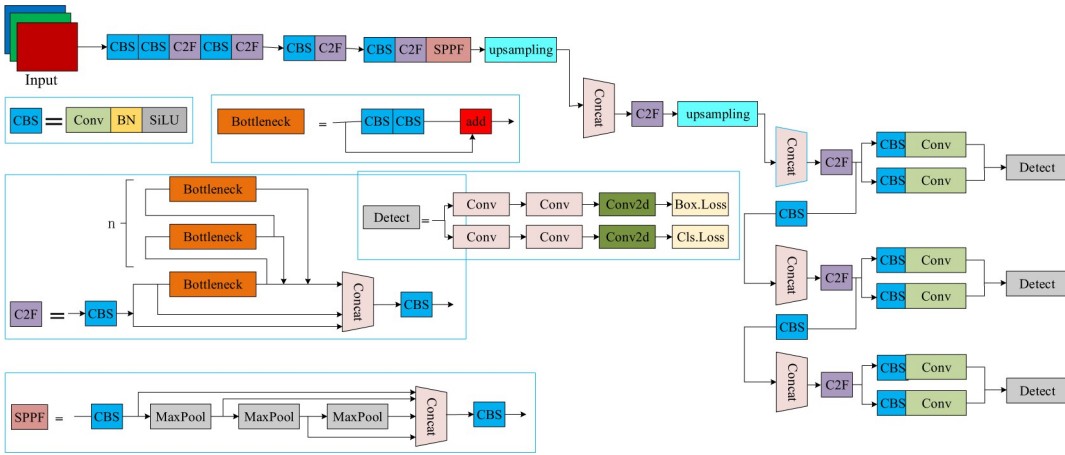

**Fig 1. Structure of the Yolov8s model.**

structure to compose the neck module. In YOLOx [32], the idea of head decoupling was used by Yolov8 for the last part of the neck. It combines confidence and regression bins to achieve different levels of accuracy. For positive and negative sample assignment, the Yolov8 algorithm uses the task-aligned allocator of TOOD to select positive samples based on the weighted scores of classification and regression. A diagram of the Yolov8s network architecture is shown in Fig 1.

## 3. Improvement of the model

The Yolov8s model is improved by replacing the Yolov8s backbone network with Fasternet and introducing Slimneck in the head module of the backbone network, which accomplishes an excellent trade-off between the accuracy and speed of the model, improves the convergence speed, reduces the number of parameters and computation of the model, and increases the detection rate. Faster convergence and more accurate regression results are obtained by using DIoU for better coal gangue differentiation. Finally, DyHead is used to significantly improve the expressive power of the object detection head of the model. The structure diagram of the improved model is shown in Fig 2.

### 3.1 BAM block

The BAMblock module is a novel attention mechanism [33]. The principle is to improve the model's ability to perceive the target by adjusting both channel attention and spatial attention to the intermediate feature maps. In channel attention, the BAMblock module weights the features of different channels by learning a vector of channel weights, while in spatial attention, a spatial weight matrix is obtained by computing the importance of each spatial location. By combining these two types of attention, the final attention map can be obtained, which leads the model to pay additional attention to the target-related features. In terms of implementation, the BAMblock module is typically integrated into a generic CNN architecture and placed at each bottleneck of the network. Multiple BAMblock modules are combined to form a hierarchical attention mechanism similar to the human perception process. Specifically, in the early stages, the BAMblock module removes low-level features. The architecture of the BAMblock module is shown in Fig 3. Given an input intermediate feature map F, the module computes the corresponding attention map M(F) by using channel attention branching (Mc) and spatial attention branching (Ms), respectively. The module has two hyperparameters: an

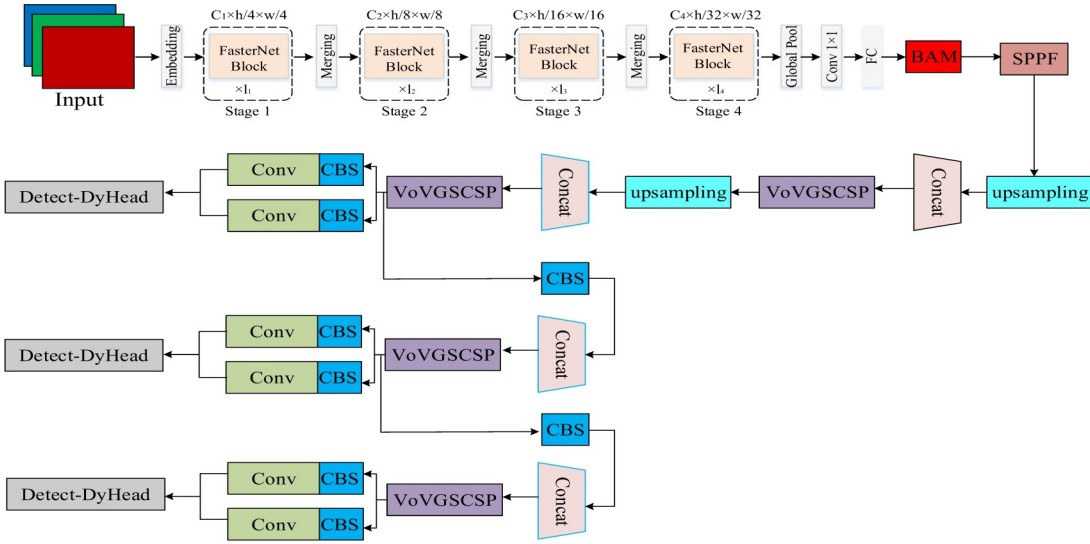

**Fig 2. Improved model structure.**

expansion value (d) and a contraction ratio (ℝ))(an expansion value set to 4, a contraction ratio set to 16). The dilation value is used to determine the receptive field size, which helps the spatial attention branch to aggregate a wider range of contextual information; Meanwhile, the shrinkage ratio controls the capacity and computational overhead of the two attention branches.

## 3.2 DIoU

The loss function is a measure of the gap between the model prediction and the true result. The loss function is often a scalar function, and a smaller value of it indicates that the model's prediction is closer to the true result, and vice versa, the larger the gap between the prediction and the true result.

When training a model using Yolov8, the parameters of the model need to be continuously adjusted to minimize the gap between the predicted and true results of the model, and the loss

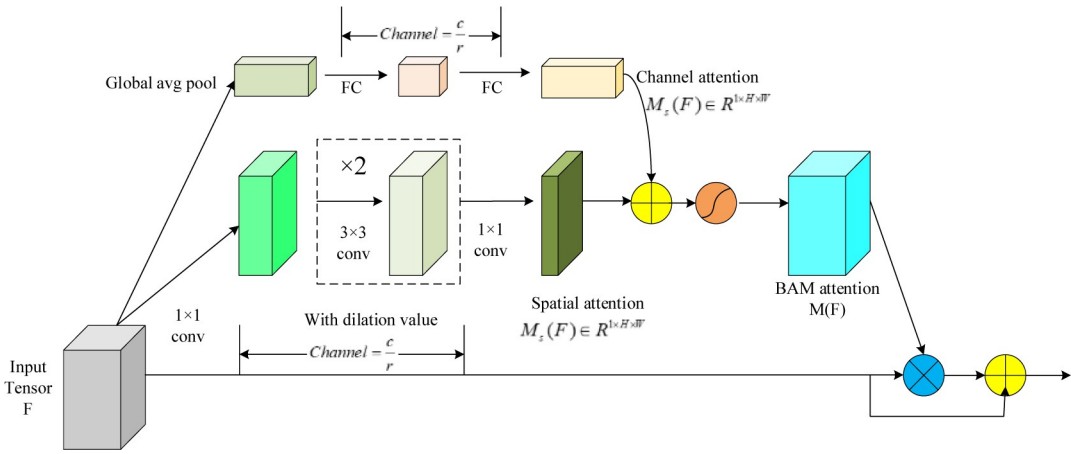

**Fig 3. BAMblock module.**

function plays a key role in this process. It measures the gap between the model prediction and the true result and converts this gap into a scalar value. The default localisation loss function used in Yolov8 is the CIoU function, which is calculated as follows [34].

$$CIoU\_Loss = 1 - IOU + \frac{\lambda^2(a, a^{gt})}{c^2} + \alpha\mu \tag{1}$$

$$\alpha = \frac{\mu}{(1 - IOU) + \mu} \tag{2}$$

$$\mu = \frac{4}{\pi}\left[(\arctan\frac{w^{gt}}{h^{gt}}) - \arctan\frac{w}{h}\right]^2 \tag{3}$$

Eqs (1)–(3) in which a and agt are the centroids of the prediction box and the target box, respectively, and λ is the Euclidean distance between the two centroids; C is the diagonal length of the minimum enclosed area of the prediction and target boxes. α is the weight of the function; μ is the consistency of the aspect ratio of the two frames; Here, h and w are the height and width of the prediction box; And hgt and wgt are the height and width of the target box, respectively. The CIoU function mainly takes care of the overlapping parts of the prediction box and the target frame.

DIoU (Distance-IoU) is an improved target frame regression method that introduces C-detection frames that take into account the relationship between true and predicted frames. Different from the traditional computation of the intersection ratio, DIoU uses Euclidean distance to measure the difference between each detection frame. This approach is more in line with the mechanism of target-box regression, and can robustly compute the distance, overlap ratio, and scale between the target and ANCHOR, making target-box regression more stable. Replacing CIoU with DIoU, the DIoU function is calculated as follows.

$$DIoU = IOU - \frac{p^2(b, b^{gt})}{c^2} = IOU - \frac{d^2}{c^2}, -1 \le DIoU \le 1 \tag{4}$$

In Eq 4, the symbol p denotes the Euclidean distance between the predicted frame b and the true target bounding box bgt. Specifically, b denotes the central coordinate parameter of the predicted bounding box and bgt denotes the central coordinate parameter of the true target bounding box. Let p2 denote the square of the distance between the two centroids, and c denote the diagonal length of the smallest outer rectangle of the two rectangles. IOU = 1 when the two frames coincide exactly, that is, the distance between the two centroids is 0 and p = 0. Thus, DIoU = 1–0 = 1, indicating that the two frames match completely. Conversely, when the two frames are extremely much apart, i.e., the distance between the two centroids is particularly large, p2/ c2 tends to be 1, which makes the IOU close to 0. DIoU = 0–1 = -1, indicating that there is no match between the two frames. Therefore, the range of values of DIoU is also [–1,1].

### 3.3 Slimneck

Slim-neck by GSConv is a method that combines channel dense convolution (SC) and channel sparse convolution (DSC) [35], where the feature mapping generated by reorganising the output channels of the DSC remains "depth-separated". In order to make the output of DSC close to that of SC, a different method, GSConv, is introduced, which employs a mixture of convolution, DSC, DSC and shuffling techniques to infiltrate the SC-generated information into the DSC-generated information. Specifically, GSConv uses a shuffling technique to uniformly

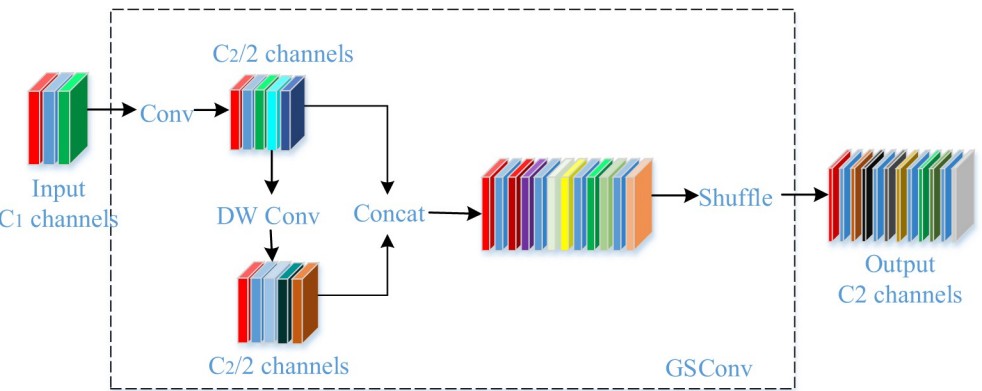

**Fig 4. GSConv module structure.**

exchange SC-generated information into each part of the DSC, thus achieving full information mixing while also reducing the computational cost and effectively exploiting the benefits of DSC.The GSConv approach employs a uniform mixing strategy that minimizes the negative impact of DSC imperfections on the model and achieves an excellent trade-off between accuracy and speed. Experimental results show that the thin-neck model using GSConv outperforms the original model in terms of accuracy and speed. Therefore, GSConv can be considered as an effective solution to overcome the DSC deficiency and improve the performance of convolutional neural networks. The structure is shown in Fig 4.

GSConv is a less computationally costly conventional operation, with a computational cost of about 50% of SC (0.5 + 0.5C1, where the larger the C1 value, the closer the ratio is to 50%). Despite the lower computational cost, the contribution of GSConv to the model learning capability is comparable to that of conventional convolution operations. To further improve the performance of GSConv, we introduce a GS bottleneck on top of GSConv, as shown in Fig 5 (A). The structure of the GS bottleneck module enhances the expressiveness of GSConv, which in turn improves the model learning. A one-time aggregation approach is used to design the cross-stage partial network (GSCSP) module, VoV-GSCSP, as shown in Fig 5(B).VoV-GSCSP provides three design alternatives, which are (b) simple, direct, and faster reasoning, and (c) and (d) design alternatives with higher reuse of these features. VoVGSCSP1 performs cost-effectively in terms of displaying better performance.

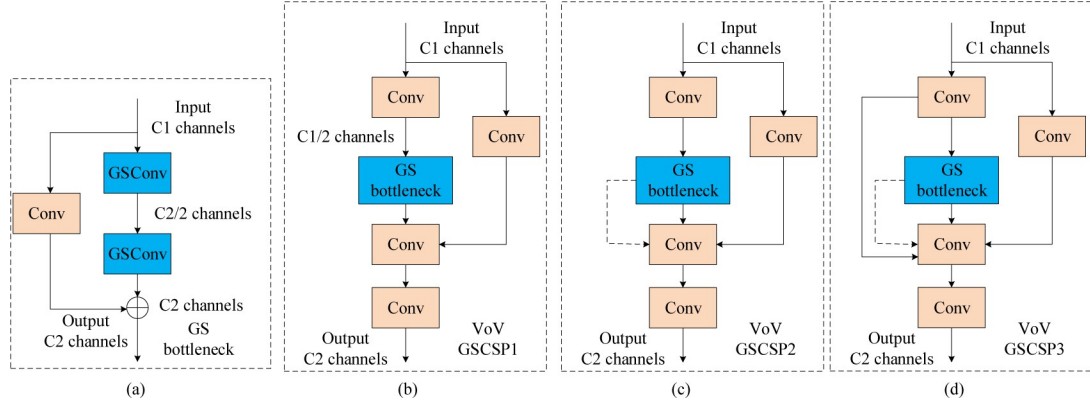

**Fig 5. VoVGSCSP module structure.**

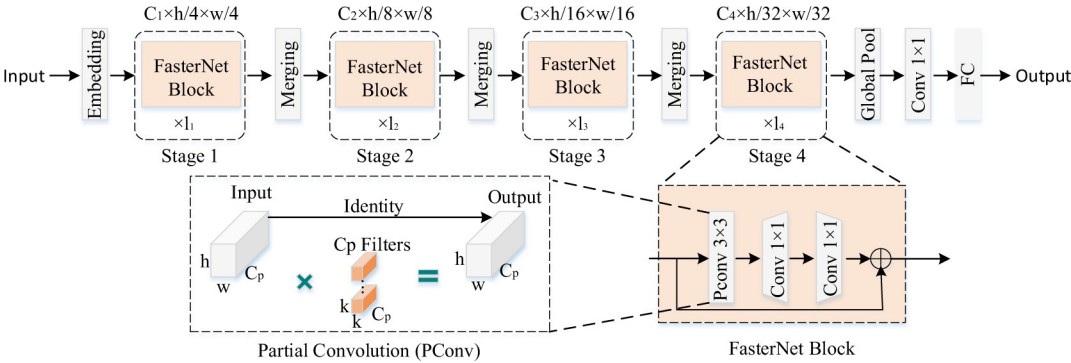

**Fig 6. Fasternet module.**

## 3.4 Fasternet

In order to construct efficient neural networks, a different partial convolution (PConv) method is proposed, which is able to extract spatial features more efficiently while reducing redundant computation and memory access. Based on the PConv technique, FasterNet, a different family of neural networks, has also been proposed. Compared to other networks, FasterNet achieves higher operating speeds on a variety of devices, which does not have an impact on the accuracy of various visual tasks [36]. FasterNet is an efficient core network built for object detection tasks and deeply optimized for speed and accuracy. The core idea of FasterNet is to optimise the representation of perceptual features and area coverage while ensuring it is lightweight and fast to run. The network architecture of FasterNet is constructed from three core components: a base network module, a fast feature fusion module, and an efficient upsampling module.

The overall framework of FasterNet is shown in Fig 6. The system consists of four stages, each of which is equipped with a set of FasterNet blocks with embedding or merging layers in front of it. The embedding layer consists of a regular Conv 4×4 with a step size of 4, while the merging layer consists of a regular Conv 2×2 with a step size of 2. Feature classification is done by the last three layers. In the internal structure of each FasterNet block, there is a PConv layer followed immediately by two PWConv layers. We add normalization and activation layers after the intermediate layers, which is done to maintain feature diversity and reduce latency.

## 3.5 DyHead

For the problem of localisation and classification in object detection, various algorithms have been improved from different perspectives, making it difficult to analyze the metrics from a unified perspective. To solve the above problems, a novel dynamic head framework, DyHead [37], is proposed, aiming to unify different target detection heads through the attention mechanism. The proposed framework achieves scale awareness through an attention mechanism between feature hierarchies, spatial awareness between spatial locations, and task awareness within the output channel to improve the expressive power of the model's object detection head without increasing the computational effort. Meanwhile, DyHead introduces dynamic convolution and adaptive sensing field mechanism to improve object detection performance. Dynamic convolution can adaptively adjust the shape of the convolutional kernel based on the size and shape of the target to better capture the feature information of the target; the detection experiments show that the introduction of DyHead can significantly improve the accuracy of the target detection algorithm in complex scenes, and it can better deal with the minor-size

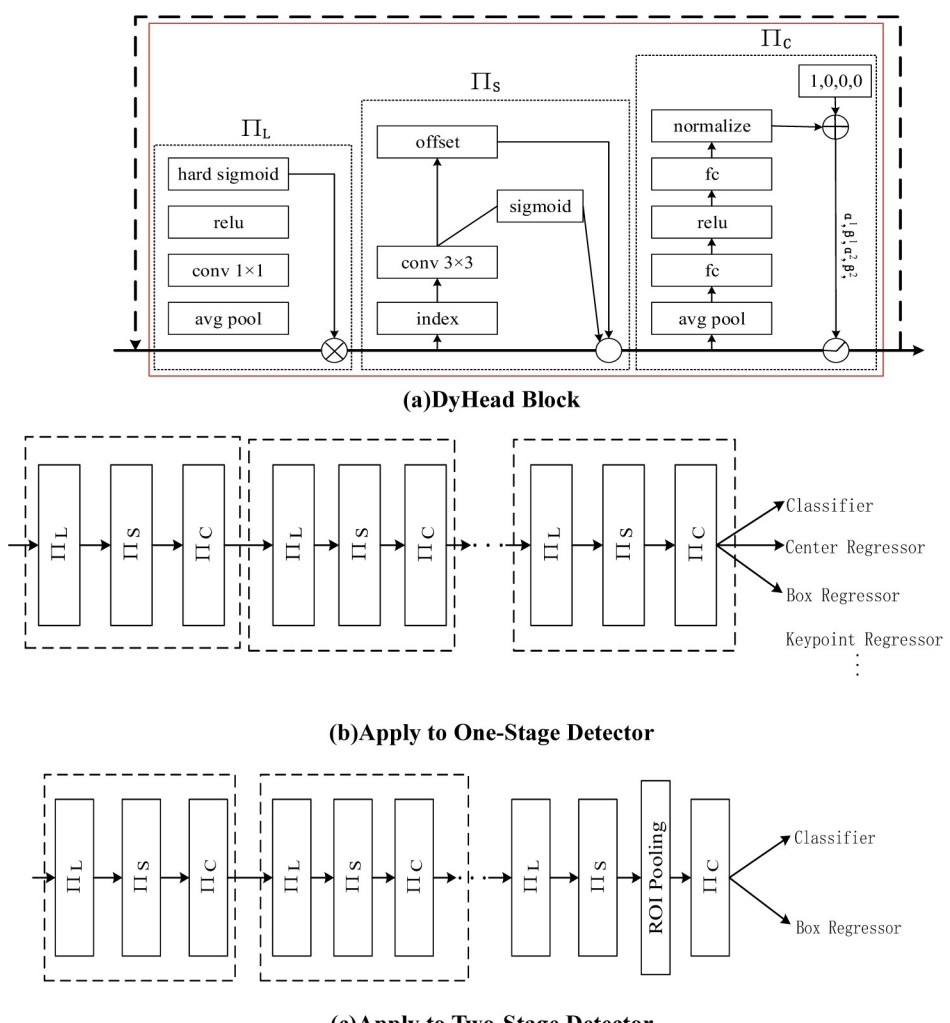

**Fig 7. Detailed design of dynamic head.** (a)DyHead Block, (b)Apply to One-Stage Detector, (c)Apply to Two-Stage Detector.

targets and the targets with large length and width comparisons, and it has a great practical value and prospects for promotion. The detailed design of the dynamic head is shown in Fig 7 (A) shows the detailed implementation of each attention block. (b) shows how to apply our dynamic head block to a one-stage target detector. (c) shows how to apply our dynamic head block to a two-level target detector.

# 4. Experiments and results

## 4.1 Data sets

Experiments were conducted to train on coal and gangue datasets. Experimentally, in order to collect image data of coal and gangue, the video was recorded in different lighting conditions with black as the background of coal and gangue, then the frames were extracted from the video and each frame was labeled by the Labelimg software. Division of training and test samples. There are 738 images in the training sample, and 128 in the test sample. In this study, we use random affine transformations as a means of data augmentation, that is, performing

random rotation, translation, rescaling, and marketing operations. This method improves the image quality and increases the amount of information by processing the image multiple times. Data augmentation using random affine functions not only improves the diversity of the data, but also helps to enhance the multiplicity of the data; On the one hand, it makes the image clearer and gives us more detailed background information. On the other hand, it effectively suppresses noise and preserves detailed information. It indirectly extends the size of Batchsize and has been successfully applied in image segmentation. The robustness and generalization ability of the model are also improved.

## 4.2 Experimental equipment and evaluation indicators

The development language for this model is mainly Python, the open-source deep learning framework PyTorch is used as the network framework, and CUDA11.3 is used to accelerate the training. The hardware test environment of this model CPU is selected as 13th Gen Intel (R) Core(TM) i5-13400F, and GPU is selected as NVIDIA's RTX3060Ti with 8G video memory. For training, the input image was set to 640×640 and the model was trained using SGD as the optimisation function. The model training period (epoch) is 200, the batch size is 16, the initial learning rate is 0.01 and the random seed seed is taken as 1.

The model evaluation metric uses mAP as the final evaluation metric, which measures the correct recognition rate of the model. GFLOPs are used to measure the complexity of a model or algorithm, while Params indicate the model size. Consequently, the smaller the Params and GFLOPs, the smaller the computational power required by the model. The number of model parameters (Params) and the amount of computation (GFLOPs) are used to measure the model complexity, and the number of frames per second (FPS) are used to measure the speed of inference. The FPS is the number of detected frames per second, and its magnitude depends not only on the weight of the algorithm, but also on the hardware configuration of the experimental equipment.

## 4.3 Experimental results

The detection results in the Yolov8s model are shown in Table 1.

The Yolov8s-D model is a replacement of the CIoU loss function, and using the DIoU loss function improves FPS by 0.92 frames/s and mAP by 0.7%, while the model size, number of parameters, and computation remain unchanged, It has been shown that DIoU improves the speed of model detection and results in better detection accuracy; The Yolov8s-DY model replaces the Detect layer with Detect-DyHead, which improves mAP by 2.1%, with a reduction of 0.6MB in model size, 0.3M in parameters, and 0.3G in computation. It is shown that the use of the Detect-DyHead module improves the detection accuracy of the model and

**Table 1. Comparison of experimental results.**

| Model name | Params/M | FLOPs/G | Model/MB | mAP/% | FPS/ (frame·s$^{-1}$) |
|---|---|---|---|---|---|
| Yolov8s | 11.1 | 28.4 | 22.5 | 91.3 | 119.48 |
| Yolov8s-D | 11.1 | 28.4 | 22.5 | 92.0 | 120.4 |
| Yolov8s-B | 11.2 | 28.5 | 22.7 | 92.2 | 96.2 |
| Yolov8s-DY | 10.8 | 28.1 | 21.9 | 93.4 | 90.9 |
| Yolov8s-S | 10.3 | 25.1 | 20.8 | 92.0 | 117.6 |
| Yolov8s-F | 8.6 | 21.7 | 17.5 | 91.0 | 107.5 |
| Yolov8s-change | 7.9 | 18.5 | 16.2 | 93.8 | 99.0 |

simultaneously reduces the model complexity and computation; The Yolov8s-B model, where the BAMblock attention mechanism is added at the last layer location in the backbone part, improves mAP by 0.9% with 0.2MB increase in model size, 0.1M increase in number of parameters, and 0.1G increase in computation. It has been shown that the use of the BAMblock attention mechanism can improve the detection accuracy of the model with a small computational effort loss; The Yolov8s-S model, a replacement for the C2F module in the head section using Slimneck, also improves mAP by 0.7% with a reduced model size of 1.7MB, a reduced number of parameters of 0.8M, and a reduced computational cost of 3.3G. It was shown that the use of this light yet point-raising method improved the data in a major way;The Yolov8s-F model, which replaces the backbone network in the Yolov8s network architecture with the lightweight network Fasternet, shows only a 0.3% reduction in mAP, a 5MB reduction in model size, a 2.5M reduction in parameter volume, and a 6.7G reduction in computation. It is shown that using Fasternet as a network can improve the detection speed and reduce the model complexity. The results show that improving the model can significantly reduce the storage size, number of parameters, and computational cost of the model and lead to improved detection accuracy.

## 4.4 Ablation experiments

In order to analyze the impact of different improvement strategies on the model detection performance, it is necessary to perform ablation experiments with several models Yolov8s, Yolov8s-D, Yolov8s-DY, Yolov8s-B, Yolov8s-S, and Yolov8s-F. The results are shown in Table 2.Among them, Yolov8s-F is a replacement of the Yolov8s backbone with Fasternet. Yolov8s-S is the replacement of C2F in the head module of Yolov8s with VoVGSCSP in the slimneck. Yolov8S-D replaces the CIoU used by default with DIoU. The Yolov8s-DY model is to replace the Detect layer with Detect-DyHead. The Yolov8s-B model is to add the BAMblock attention mechanism at the last layer position in the backbone profile. The improved model uses a more efficient network structure compared to the network structure of Yolov8s, which improves the accuracy and reduces the number of parameters and computational cost of the model. It is also demonstrated that the VoVGSCSP module does not reduce the accuracy of the algorithm, but rather reduces the number of parameters and computation of the model. The use of the Fasternet backbone network reduces the computation and number of parameters. However, the use of Detect-DyHead effectively improves the detection accuracy. The use of DIoU with the BAMblock attention mechanism better combines the above improvements with the Yolov8n algorithm. The improved model reduces the model storage size, the number of model parameters, and the amount of computation by 28, 28.8, and 34.8%, respectively. The detection accuracy was also improved by 2.5%. This effectively reduces the difficulty and cost of deploying the model at mobile terminals, and enables the algorithm to meet real-time performance while achieving a considerable increase in accuracy.

**Table 2. Comparison of ablation experiments.**

| Yolov8s | Yolov8s-D | Yolov8s-DY | Yolov8s-B | Yolov8s-S | Yolov8s-F | Model/MB | mAP/% |
|---|---|---|---|---|---|---|---|
| √ | | | | | | 22.5 | 91.3 |
| √ | | | | √ | | 20.8 | 91.9 |
| √ | | | | √ | √ | 16.5 | 92.6 |
| √ | | √ | | √ | √ | 16.0 | 93.2 |
| √ | √ | √ | | √ | √ | 16.0 | 92.9 |
| √ | √ | √ | √ | √ | √ | 16.2 | 93.8 |

**Table 3. Comparison of results of different algorithms.**

| Model name | Params/M | FLOPs/G | Model/MB | mAP/% |
|---|---|---|---|---|
| Yolov3-tiny | 12.1 | 19.0 | 24.3 | 91.1 |
| Yolov4 | 64 | 29.9 | 244 | 77.13 |
| Yolov5s | 9.1 | 23.8 | 18.5 | 92.2 |
| Yolov6s | 16.2 | 44.0 | 32.8 | 92.1 |
| Yolov7-tiny | 6.0 | 13.2 | 12.3 | 90.8 |
| Yolov8n | 3.0 | 8.1 | 6.2 | 91.1 |
| Yolov8n-change | 4.5 | 11.5 | 9.4 | 92.2 |
| Yolov8s | 11.1 | 28.4 | 22.5 | 91.3 |
| Yolov8s-change | 7.9 | 18.5 | 16.2 | 93.8 |

## 4.5 Comparison of results of different algorithmic models

Different versions of four mainstream object detection models, Yolov3, Yolov4, Yolov5, Yolov6, Yolov7, and Yolov8, were selected for comparison, the hyperparameters and training parameters of the models were set to their default values and the datasets were used for training. Yolov3, Yolov5, Yolov6 are in the Yolov8 source code. Yolov4, Yolov7 are in the official source code. The results are shown in Table 3 and show that the improved model is effective on different versions of Yolov8, and the improvement on the Yolov8n model improves the detection accuracy by 1.1%, but the model size, number of parameters, and computational effort become cumbersome. The improvement of the Yolov8n model increases the detection accuracy by 2.5%, while the model size, number of parameters, and computation are optimized to take into account the detection accuracy and detection rate. The improved Yolov8s model has the best detection accuracy mAP among these models, which demonstrates the superiority of the proposed improved model.

## 4.6 Visualisation and analysis of test results

In order to evaluate the performance of the proposed improved algorithm more intuitively, the detection performance of Yolov8s and the proposed improved model of Yolov8s is demonstrated on a test set of graphs, where both models are able to properly detect and distinguish coal gangue. As shown in Fig 8, coal is detected in red and gangue in pink. (a) The group because of the light problem, resulting in the original model misdetection, while the improved model has a stronger ability to suppress interference and has better anti-interference ability, indicating that the improved model has a stronger context-awareness and feature fusion ability. Group (b) pictures because of the background problem, the original model did not detect this little part of gangue, while the improved model can locate it more fully; By enhancing the model's perception and understanding of contextual information and feature fusion capabilities, and mitigating the interference of background factors, the improved model is able to localize and identify defects more accurately and improve detection accuracy. The experiment gives a comparison of the loss curves of Yolov8s and Yolov8s-change bounding box, as shown in Fig 9, it can be seen that the loss values of the two algorithms are rapidly decreasing in the first 25 rounds, and the loss curves of Yolov8s-change converge faster than those of the Yolov8s algorithm with better convergence, which indicates that the improved algorithm has a stronger learning ability and a better detection effect for the detection of experimental targets. The training accuracy curves of Yolov8s and Yolov8s-change models are shown in Fig 10, which shows that the accuracy of the improved algorithm is lower in the pre-training period, and the accuracy curve is basically the same as that of the original algorithm after about 75

Yolov8s

Modelling

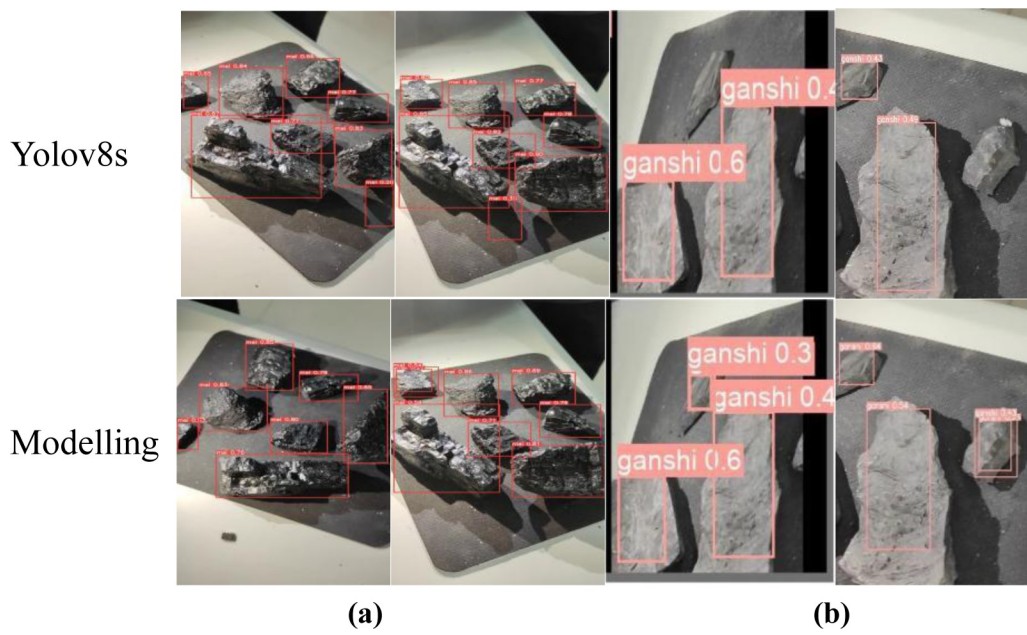

(a) (b)

**Fig 8. Comparison of detection results.**

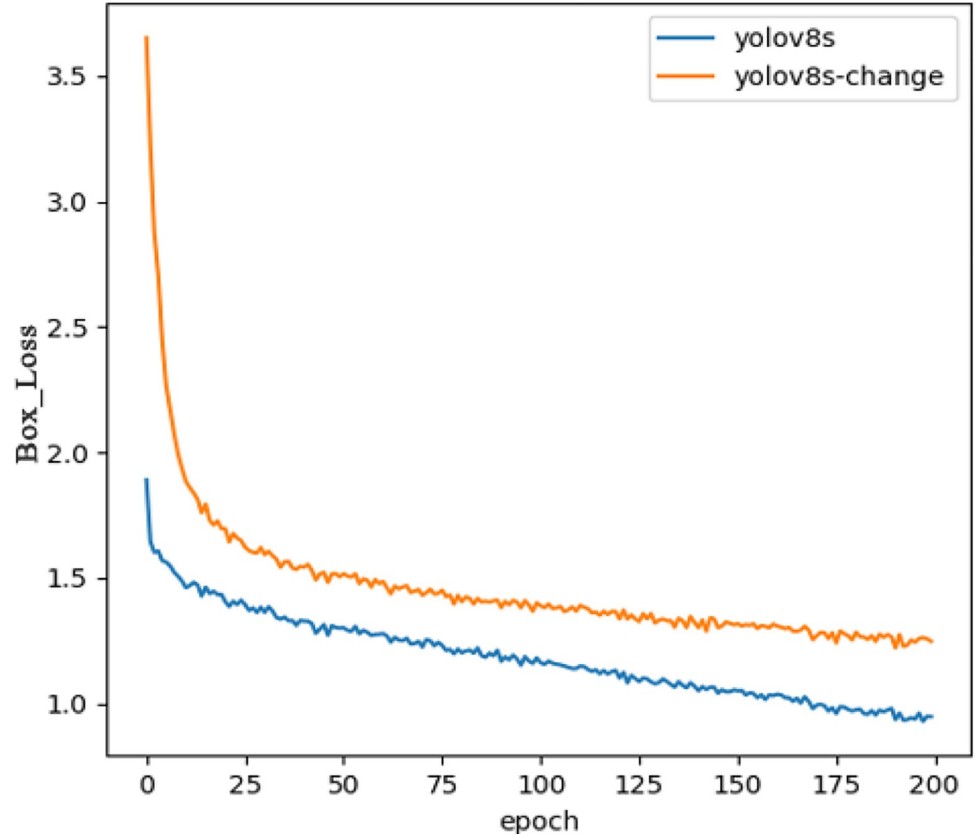

**Fig 9. Comparison of boundary box loss curves.**

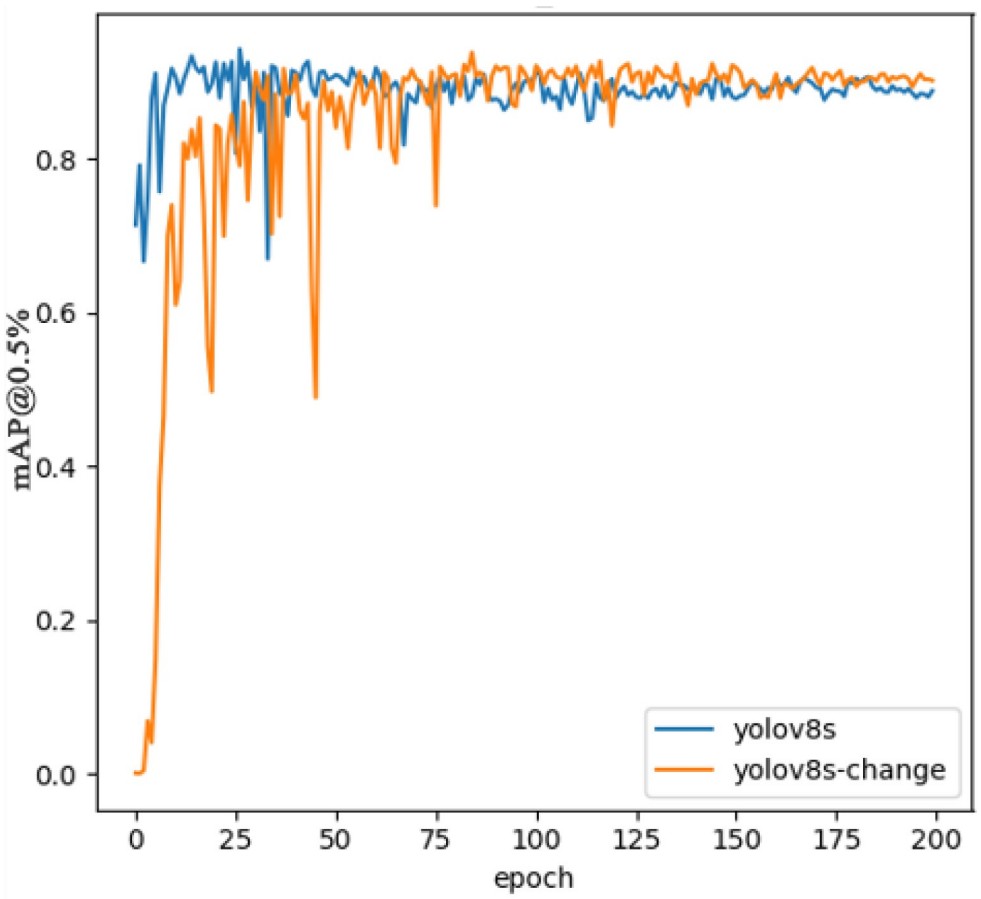

**Fig 10. Comparison of accuracy curves.**

iterations of training, and the accuracy of Yolov8s-change is significantly higher than that of the original algorithm in the end.

## 5. Conclusion

The proposed modified Yolov8s based method for gang recognition solves the problem of traditional Yolov8s applied to object detection. The Slimneck structure comprehensively extracts the global and local feature information of gang image, which improves the gang detection accuracy and reduces the computational effort and number of parameters. The BAMblock attention mechanism can significantly improve the detection accuracy of the model while increasing the computational cost by only a small amount. The Fasternet backbone network can considerably reduce the amount of computation and the number of parameters with only a 0.3% loss in accuracy. The Detect-DyHead module is introduced in the Yolov8s network, which not only compresses the number of model parameters but also improves the detection accuracy. It has comparable or better detection performance than other algorithms.

Experiments show that the improved model has the advantage of fewer parameters, lower computational cost, higher detection accuracy, and meets the real-time requirements. The improved model compresses the storage size of the model by 28%, reduces the amount of parameters by 28.8%, reduces the amount of computation by 34.8%, and improves the detection accuracy by 2.5% compared to the original model. Compared to existing models, the

proposed method achieves higher detection accuracy while reducing the requirements on the computational and storage capacity of the platform, and is easy to deploy on resource-constrained devices.

## Author Contributions

**Conceptualization:** Zhibo Fu, Zhengkun Xie, RunZhi Li.

**Data curation:** Zhibo Fu, Xinpeng Yuan.

**Formal analysis:** Zhibo Fu, Xinpeng Yuan.

**Funding acquisition:** Xinpeng Yuan.

**Investigation:** Zhengkun Xie, Li Huang.

**Methodology:** Zhibo Fu.

**Project administration:** Zhibo Fu, RunZhi Li.

**Resources:** Zhibo Fu.

**Software:** RunZhi Li.

**Supervision:** Zhengkun Xie, Li Huang.

**Validation:** Xinpeng Yuan, RunZhi Li.

**Visualization:** Li Huang.

**Writing – original draft:** Zhibo Fu.

**Writing – review & editing:** Xinpeng Yuan.

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
