## [Decision Letter · Decision Letter 0]

12 Dec 2023

PONE-D-23-34109Research on coal gangue detection algorithm with improved Yolov8sPLOS ONE

Dear Dr. FU,

Thank you for submitting your manuscript to PLOS ONE. After careful consideration, we feel that it has merit but does not fully meet PLOS ONE’s publication criteria as it currently stands. Therefore, we invite you to submit a revised version of the manuscript that addresses the points raised during the review process.

We look forward to receiving your revised manuscript.

Kind regards,

Khalil Abdelrazek Khalil, Ph.D.

Academic Editor

PLOS ONE

DC-YOLOv8: Small-Size Object Detection Algorithm Based on Camera Sensor - https://doi.org/10.3390/electronics12102323

Run, Don’t Walk: Chasing Higher FLOPS for Faster Neural Networks - http://export.arxiv.org/pdf/2303.03667

In your revision ensure you cite all your sources (including your own works), and quote or rephrase any duplicated text outside the methods section. Further consideration is dependent on these concerns being addressed.

“This research was funded by the Scientific and Technological Innovation Project of Colleges and Universities in Shanxi Province (2019-768); This research was funded by a grant from the Teaching Reform and Innovation Project of Higher Education Institutions in Shanxi Province (General Project) (J20220865); This research was funded by a grant from the Shanxi Datong University Graduate Education Reform Research Program(21JG17); This research was funded by the Graduate Education Innovation Project of Shanxi Datong University (23CX49,23CX50).”

“1.Science and Technology Innovation Program for Higher Education Institutions in Shanxi Province

2.Teaching Reform and Innovation Program for Higher Education Institutions in Shanxi Province (General Program)

3.Shanxi Datong University Graduate Education Reform Research Program

4.Shanxi Datong University Graduate Education Innovation Program”

7. In your Data Availability statement, you have not specified where the minimal data set underlying the results described in your manuscript can be found. PLOS defines a study's minimal data set as the underlying data used to reach the conclusions drawn in the manuscript and any additional data required to replicate the reported study findings in their entirety. All PLOS journals require that the minimal data set be made fully available. For more information about our data policy, please see http://journals.plos.org/plosone/s/data-availability.

Reviewers' comments:

Reviewer's Responses to Questions

**Comments to the Author**

1. Is the manuscript technically sound, and do the data support the conclusions?

Reviewer #1: Yes

Reviewer #2: Partly

Reviewer #3: Partly

2. Has the statistical analysis been performed appropriately and rigorously? 

Reviewer #1: Yes

Reviewer #2: Yes

Reviewer #3: No

3. Have the authors made all data underlying the findings in their manuscript fully available?

Reviewer #1: Yes

Reviewer #2: Yes

Reviewer #3: No

4. Is the manuscript presented in an intelligible fashion and written in standard English?

Reviewer #1: No

Reviewer #2: Yes

Reviewer #3: No

5. Review Comments to the Author

Reviewer #1: 1. Please standardize the use of English, the international language, wherever Chinese appears in fig.1,fig.2,table1 and table 3.

2. The presence of “The Yolov8s model is improved by replacing the Yolov5s backbone network with Fasternet” in2. “Improvement of the model” is puzzling.

3. Fig.5(c) (d) No specific difference between the two structures in the figure, please add in the text or in the figure.

4. 2.5 DyHead Please provide additional information in the form of a formula or diagram.

5. 3.1 Data sets indicate only the number of coal and gangue images, not the number of target boxes.

6. 3.1 Data sets Coal and gangue quantities are not consistent, is there an imbalance in the data categories? Please add more metrics, e.g. F1-Score.

7. In real scenes, images may be affected by different lighting, occlusion, and other factors. The dataset does not take occlusion into account.

8. Please supplement the Network's detection results for "In different lighting conditions" in 3.1 Data sets.

9. In fig.7, labels are confusing.

10. In fig.7, the image is suspected to be data-enhanced, but it is not accounted for in the paper.

11. How are the two parameters expansion value and a contraction ratio set in "2.1 BAMblock"?

12. What is the structure of the merging module in "Fig. 6 Fasternet module"? Add a description in the paper.

13. Why did the Yolov8s-B model add the BAM attention mechanism only at the last layer of the backbone?

14. fig.1 Structural error of Yolov8s in the figure.

15. Percent should be replaced by % throughout the paper.

16. Why did the improvements in the text not work well with Yolov8n?

17. Yolov4 is not included in the table 3 model comparison?

18. The caption “2.4 Fanternet” network name errors.

19. The training loss process for the improved Yolov8s target detection network is not shown, e.g., the loss curve is shown.

20. "Coal gang target detection" is misspelled in the abstract.

Reviewer #2: This paper utilize YoloV8 to improve coal gangue detection task. The authors apply several deep learning methods to enhance the model performance, facilitate training as well as inference speed, and reduce the model size.

Some questions about this paper:

1. How did the dataset used this paper collect? It is not clear how the data collection and annotation process are. How was the data preprocessing?

2. I wonder how the hyperparameters are identified. Did the author have hyperparameter searching?

3. In Table 1, what are D, B, DY, S stand for? Please explain the abbreviations in the caption.

4. Table 1 have a Chinese character. Please fix it.

5. Please cite YoloV8 at first you mention it.

6. Please have a white space before a new sentence. You lost many white space in the paper. E.g., in page 2, [9].The

7. 2.1 BAMblock  2.1 BAM block

8. 2.2 Diou  2.2 DIoU. Please use the capitalization consistently.

Reviewer #3: 1. The title and abstract in the manuscript only reflect the algorithm model of coal gangue detection, and do not demonstrate the technical means and description of detection. Please add it to the list, authors.

2. The overall process of this study needs to be reflected in the last paragraph of the article introduction.

3. The formula needs to be displayed in the center.

4. The author provides more descriptions of algorithm principles, while there is less analysis of experiments and results, requiring optimization of the description.

5. There are 738 images in the training sample, of which 500 are coal and 238 are gang, and 128 in the test sample, of which 78 are coal and 50 are gang Can the final test results of this manuscript ensure reliability, as there is a significant difference in the quantity of coal and gangue and the samples are imbalanced? Please explain them.

6. Why does Chinese appear in Table 3? Please modify it.

7. More quantitative indicators need to be reflected in the Conclusion to ensure the persuasiveness of the manuscript.

8. What are the shortcomings and areas that need to be improved in this manuscript.

9. Inconsistent reference format.

10. To ensure the convenience of researchers in this field, authors need to upload the data and related code to the Github.

6. PLOS authors have the option to publish the peer review history of their article (what does this mean?). If published, this will include your full peer review and any attached files.

Reviewer #1: No

Reviewer #2: No

Reviewer #3: No

---

## [Author Response · Author response to Decision Letter 0]

18 Jan 2024

Response to Reviewer 1

Thank you for your review of our paper. We have answered each of your points below.

1. [Please standardize the use of English, the international language, wherever Chinese appears in fig.1,fig.2,table1 and table 3.]

Response: In Figures 1 and 2, Tables 1 and 3, the Chinese characters have been changed to English wherever they appear.

2. [The presence of “The Yolov8s model is improved by replacing the Yolov5s backbone network with Fasternet” in2. “Improvement of the model” is puzzling.]

Response: The problem was a clerical error that has been corrected.(p.4, Sixth line from the end)

3. [Fig.5(c) (d) No specific difference between the two structures in the figure, please add in the text or in the figure.]

Response: The problem was a problem with the drawing of the diagram, which has been modified. (p.12, fig. 5)

4. [2.5 DyHead Please provide additional information in the form of a formula or diagram.]

Response: Added graphic descriptions to the DyHead section. (p.14, fig. 7 and yellow marker text)

5. [3.1 Data sets indicate only the number of coal and gangue images, not the number of target boxes.]

Response: Changes have been made. The number of target boxes has been removed. (p.15, lines 4-5)

6. [3.1 Data sets Coal and gangue quantities are not consistent, is there an imbalance in the data categories? Please add more metrics, e.g. F1-Score.]

Response: It is true that there are high and low category accuracies, but category accuracies are not necessarily the result of sample imbalance, and the accuracy of the model ensures the reliability of the experiment. I trained after expanding the dataset by rotating, translating, scaling, and flipping horizontally to make the numbers consistent, and this is the changed F1-Score.

7. [In real scenes, images may be affected by different lighting, occlusion, and other factors. The dataset does not take occlusion into account.]

Response: On the issue of dataset occlusion during training, I am treating the detection targets of incomplete datasets as occlusion for the purpose of detection. I am not sure if this is occlusion. Next time I will do Cutout data enhancement for the occlusion problem.Cutout is a new regularisation method.Cutout is a new regularisation method which randomly subtracts a portion of the image during the training process. Some similar occlusion objects are generated by cutting.

8. [Please supplement the Network's detection results for "In different lighting conditions" in 3.1 Data sets.]

Response: Different lighting conditions were used because the dataset was produced by video-recording the coal and gangue and then sampling the footage and using lights to illuminate the samples in different directions during the video-recording.

9. [In fig.7, labels are confusing.]

Response: The labelling confusion in figure 7 has been modified. (p.20, fig. 8)

10. [In fig.7, the image is suspected to be data-enhanced, but it is not accounted for in the paper.]

Response: In this paper, we use the Random affine data enhancement method to enhance the data; that is, random rotation, translation, scaling, and miscutting operations. The use of Random affine data enhancement, on the one hand, enhances the diversity of the data; provides richer contextual information; increases the Batchsize in disguise; and improves the robustness and generalisation ability of the model. It has been added in the dataset introduction section. (p.15, lines 5-13)

11. [How are the two parameters expansion value and a contraction ratio set in "2.1 BAMblock"?]

Response: In the paper BAM: Bottleneck Attention Module experimentally determined the expansion and contraction ratios, with the expansion value determining the size of the receptive field in the spatial attention branch. Performance improves as the expansion value increases, although it saturates at an expansion value of 4. The standard convolution (i.e., an expansion value of 1) produced the lowest accuracy. The shrinkage ratio is directly related to the number of channels in the two attention branches, which allows us to control the capacity and overhead of our module. a shrinkage ratio of 16 achieves the best accuracy, although the reduction ratios of 4 and 8 have a higher capacity. This result is overfitting because the training loss converges in both cases. We set the expansion value to 4 and the contraction ratio to 16 in our experiments. (p.8, fifthly line from the end)

12. [What is the structure of the merging module in "Fig. 6 Fasternet module"? Add a description in the paper.]

Response: In accordance with your suggestion it has been described in the paper. (p.13, lines 3-4)

13. [Why did the Yolov8s-B model add the BAM attention mechanism only at the last layer of the backbone?]

Response: Adding the BAM attention mechanism to the last layer of the trunk gave the best results, so I just did it here.

14. [fig.1 Structural error of Yolov8s in the figure.]

Response: The structural errors in Figure 1 have been modified. Changes to the C2F structure section. (p.7, fig.1 )

15. [Percent should be replaced by % throughout the paper.]

Response: Throughout the paper, the use of % instead of percent has been modified.

16. [Why did the improvements in the text not work well with Yolov8n?]

Response: The fastenet method in the improved model in this paper, when applied to yolov8n, increases the number of parameters, the amount of computation and the size of the model, resulting in the entire improved method also increasing the number of parameters, the amount of computation and the size of the model for yolov8n.

17. [Yolov4 is not included in the table 3 model comparison?]

Response: Yolov3-tiny, yolov5, yolov6 in the paper are the models in the yolov8 source file, which are yolov3 in yolov8, yolov5 in yolov8, yolov6 in yolov8, so the effect of the yolov4 model was not compared, and now it has been made to add and explain to the paper. Yolov3-tiny,yolov5,yolov6 are explained.(p.19,lines 1-2 and table 3)

18. [The caption “2.4 Fanternet” network name errors.]

Response: A change has been made to the title "2.4 Fanternet" network name error. (p.12,lines 6)

19. [The training loss process for the improved Yolov8s target detection network is not shown, e.g., the loss curve is shown.]

Response: Section 3.6 adds graphical and textual supplements to the graphical comparison of the bounding box loss curves before and after the improvements. (p.20,p.21,lines 5-10 and fig.9)

20. ["Coal gang target detection" is misspelled in the abstract.]

Response: The problem of spelling errors has been corrected. (p.1, lines 2)

Response to Reviewer 2

Thank you for your review of our paper. We have answered each of your points below.

1. [How did the dataset used this paper collect? It is not clear how the data collection and annotation process are. How was the data preprocessing?]

Response: Coal and gangue are extracted from a mine and then the video recording is done, then the video is sampled for frames, and then it is selected, the blurred pictures are removed, and then the pictures of each one are labelled by Labelimg software. Division of training samples and test samples. In this paper, Random affine affine transform data enhancement method is used for enhancement; i.e., random rotation, translation, scaling, and miscut operations. The use of Random affine data enhancement, on the one hand, can enhance the diversity of the data; provide richer contextual information; increase the Batchsize in disguise; and improve the robustness and generalisation ability of the model. Fix the image size in the dataset to 640×640. In this paper, we use the Random affine data enhancement method to enhance the data; that is, random rotation, translation, scaling, and miscutting operations. The use of Random affine data enhancement, on the one hand, enhances the diversity of the data; provides richer contextual information; increases the Batchsize in disguise; and improves the robustness and generalisation ability of the model.

2. [I wonder how the hyperparameters are identified. Did the author have hyperparameter searching?]

Response: My graphics card is rtx3060, it consumes less power and is slower with smaller values of workers, while it consumes more power and is faster with larger values of workers. For batch-size, it is found to be more efficient when set to a multiple of 8. At smaller values of workers, the graphics card computational resources are not fully utilised, resulting in lower power consumption and slower speed. At larger workers values, the graphics card's computing resources are fully utilised, resulting in higher power consumption and faster speeds. Setting the workers to 4 and setting the batch-size to a multiple of 8 makes it possible to fully utilise the graphics memory and achieve higher efficiency. These adjustments may also be affected by other factors such as the size of the dataset and the requirements of the task, so several experiments and adjustments may be needed to find the best combination of hyperparameters. Other hyperparameters use the default parameters officially used by yolov8.

3. [In Table 1, what are D, B, DY, S stand for? Please explain the abbreviations in the caption.]

Response: he Yolov8s-D model in Table 1 is to replace the CIOU loss function and use the Diou loss function, adopting the first letter D of Diou for Yolov8s-D. The Yolov8s-B model is to add the BAM attentional mechanism at the last layer position in the backbone section, adopting the first letter B of BAM for Yolov8s-B. The Yolov8s-DY model is to replace the Detect layer with Detect-DyHead, adopting the first two letters Dy of DyHead as Yolov8s-Dy. The Yolov8s-S model is to use Slimneck to replace the C2F module of the head section, adopting the first letter S of Slimneck as Yolov8s- S. The Yolov8s-F model is the replacement of the backbone network in the Yolov8s network structure with the lightweight network Fasternet, using the initial F of Fasternet for Yolov8s-F.

4. [Table 1 have a Chinese character. Please fix it.]

Response: The Chinese-language questions in table 1 have been amended to regularise the use of English.(p.16, table 1)

5. [Please cite YoloV8 at first you mention it.]

Response: At the time I wrote this article I had not found any yolov8 papers related to gangue recognition, so I did not cite them. Now I have also found only one and have added it.(p.3, lines 9)

6. [Please have a white space before a new sentence. You lost many white space in the paper. E.g., in page 2, [9].The]

Response: The issue of spaces before new sentences throughout the text has been modified.

7. [2.1 BAMblock  2.1 BAM block]

Response: Changes have been made to the BAMblock.

8. [2.2 Diou  2.2 DIoU. Please use the capitalization consistently.]

Response: Changes have been made to address the issue of capitalisation and consistency throughout the text.

Response to Reviewer 3

Thank you for your review of our paper. We have answered each of your points below.

1. [The title and abstract in the manuscript only reflect the algorithm model of coal gangue detection, and do not demonstrate the technical means and description of detection. Please add it to the list, authors.]

Response: Changes have been made to the title and abstract based on your suggestions.

2. [The overall process of this study needs to be reflected in the last paragraph of the article introduction.]

Response: The overall process of this study is added at the end of the introduction to the article. (p.5, lines 1-4)

3. [The formula needs to be displayed in the center.]

Response: Changes have been made to the centring of formulas. (p.9,p.10, formula 1-4)

4. [The author provides more descriptions of algorithm principles, while there is less analysis of experiments and results, requiring optimization of the description.]

Response: Section 3.6 adds a description of the graphical comparison of the description of the loss curves of the bounding box before and after the improvement. (p.20,p.21, lines 10-14 and fig.10)

5. [There are 738 images in the training sample, of which 500 are coal and 238 are gang, and 128 in the test sample, of which 78 are coal and 50 are gang Can the final test results of this manuscript ensure reliability, as there is a significant difference in the quantity of coal and gangue and the samples are imbalanced? Please explain them.]

Response: The accuracy of the model can be the best proof, according to your suggestion, I made the training sample of coal and gangue images by rotating, translating, scaling, horizontal flipping methods to expand the data set to achieve the same number of training, and the results are very small difference with the original data set.

6. [Why does Chinese appear in Table 3? Please modify it.]

Response: The Chinese questions in table 3 have been amended in English. (p.19, table 3)

7. [More quantitative indicators need to be reflected in the Conclusion to ensure the persuasiveness of the manuscript.]

Response: Changes have been made by adding specific quantitative indicators as you suggested. (p.22, lines 1-4)

8. [What are the shortcomings and areas that need to be improved in this manuscript.]

Response: There may be common problems with homemade datasets, and there may be errors in the linguistic presentation of this paper, and it is also difficult for identification techniques to achieve field application.

9. [Inconsistent reference format.]

Response: Formatting of references has been modified using the GB/T 7714 harmonised format. (p.22,p.23 References)

10. [To ensure the convenience of researchers in this field, authors need to upload the data and related code to the Github.]

Response: The relevant data have been uploaded to Github.zhibofu (github.com).

Responses to reviewers have been uploaded as a separate document

---

## [Decision Letter · Decision Letter 1]

3 Mar 2024

PONE-D-23-34109R1Research on improved gangue target detection algorithm based on Yolov8sPLOS ONE

Dear Dr. FU,

Thank you for submitting your manuscript to PLOS ONE. After careful consideration, we feel that it has merit but does not fully meet PLOS ONE’s publication criteria as it currently stands. Therefore, we invite you to submit a revised version of the manuscript that addresses the points raised during the review process.

We look forward to receiving your revised manuscript.

Kind regards,

Khalil Abdelrazek Khalil, Ph.D.

Academic Editor

PLOS ONE

Journal Requirements:

Reviewers' comments:

Reviewer's Responses to Questions

**Comments to the Author**

1. If the authors have adequately addressed your comments raised in a previous round of review and you feel that this manuscript is now acceptable for publication, you may indicate that here to bypass the “Comments to the Author” section, enter your conflict of interest statement in the “Confidential to Editor” section, and submit your "Accept" recommendation.

Reviewer #1: All comments have been addressed

Reviewer #2: All comments have been addressed

2. Is the manuscript technically sound, and do the data support the conclusions?

Reviewer #1: Yes

Reviewer #2: Yes

3. Has the statistical analysis been performed appropriately and rigorously? 

Reviewer #1: Yes

Reviewer #2: No

4. Have the authors made all data underlying the findings in their manuscript fully available?

Reviewer #1: No

Reviewer #2: No

5. Is the manuscript presented in an intelligible fashion and written in standard English?

Reviewer #1: Yes

Reviewer #2: Yes

6. Review Comments to the Author

Reviewer #1: The author has solved the problems in the manuscript and improved the quality of the manuscript, and it can be considered for acceptance.

Reviewer #2: Thanks for addressing the issues in your new version.

I wonder why the new version has much more line space than last version. Please ensure you are using correct template.

Section: 3.1: You should clarify who are the annotators, do you have annotation agreement between annotators?

7. PLOS authors have the option to publish the peer review history of their article (what does this mean?). If published, this will include your full peer review and any attached files.

Reviewer #1: No

Reviewer #2: No

---

## [Author Response · Author response to Decision Letter 1]

6 Mar 2024

Response to Reviewer 1

Thank you for your review of our paper.

Response to Reviewer 2

Thank you for your review of our paper. We have answered each of your points below.

1. [I wonder why the new version has much more line space than last version. Please ensure you are using correct template.]

Response: I found in the submission instructions "The manuscript text should be double-spaced", so the correct template is being used. The manuscript text is double-spaced.

Response to Reviewer 2

Thank you for your review of our paper. We have answered each of your points below.

1. [You should clarify who are the annotators, do you have annotation agreement between annotators?]

Response: I'm sorry maybe I didn't think through your question, the dataset question was done by me at the time, there were no other annotators and no other disagreements between authors.

---

## [Decision Letter · Decision Letter 2]

27 Mar 2024

Research on improved gangue target detection algorithm based on Yolov8s

PONE-D-23-34109R2

Dear Dr. FU,

We’re pleased to inform you that your manuscript has been judged scientifically suitable for publication and will be formally accepted for publication once it meets all outstanding technical requirements.

Kind regards,

Khalil Abdelrazek Khalil, Ph.D.

Academic Editor

PLOS ONE

Additional Editor Comments (optional):

Reviewers' comments:

Reviewer's Responses to Questions

**Comments to the Author**

1. If the authors have adequately addressed your comments raised in a previous round of review and you feel that this manuscript is now acceptable for publication, you may indicate that here to bypass the “Comments to the Author” section, enter your conflict of interest statement in the “Confidential to Editor” section, and submit your "Accept" recommendation.

Reviewer #2: All comments have been addressed

2. Is the manuscript technically sound, and do the data support the conclusions?

Reviewer #2: Partly

3. Has the statistical analysis been performed appropriately and rigorously? 

Reviewer #2: Yes

4. Have the authors made all data underlying the findings in their manuscript fully available?

Reviewer #2: Yes

5. Is the manuscript presented in an intelligible fashion and written in standard English?

Reviewer #2: Yes

6. Review Comments to the Author

Reviewer #2: The paper has addressed my concerns. I suggest to accept the paper. For the future work, I hope that the author can enhance the annotation process by involving more annotators.

7. PLOS authors have the option to publish the peer review history of their article (what does this mean?). If published, this will include your full peer review and any attached files.

Reviewer #2: No

---

## [Editor Report · Acceptance letter]

4 Apr 2024

PONE-D-23-34109R2 

PLOS ONE

Dear Dr. Fu, 

I'm pleased to inform you that your manuscript has been deemed suitable for publication in PLOS ONE. Congratulations! Your manuscript is now being handed over to our production team.

Kind regards, 

on behalf of

Dr. Khalil Abdelrazek Khalil 

Academic Editor

PLOS ONE